# Immunomodulatory Potential of Fungal Extracellular Vesicles: Insights for Therapeutic Applications

**DOI:** 10.3390/biom13101487

**Published:** 2023-10-06

**Authors:** Stefano Nenciarini, Duccio Cavalieri

**Affiliations:** Department of Biology, University of Florence, Via Madonna del Piano 6, Sesto Fiorentino, 50019 Florence, Italy; stefano.nenciarini@unifi.it

**Keywords:** extracellular vesicles, fungi, yeast, immunity, therapy, host-microbe interaction, drug delivery, vaccine

## Abstract

Extracellular vesicles (EVs) are membranous vesicular organelles that perform a variety of biological functions including cell communication across different biological kingdoms. EVs of mammals and, to a lesser extent, bacteria have been deeply studied over the years, whereas investigations of fungal EVs are still in their infancy. Fungi, encompassing both yeast and filamentous forms, are increasingly recognized for their production of extracellular vesicles (EVs) containing a wealth of proteins, lipids, and nucleic acids. These EVs play pivotal roles in orchestrating fungal communities, bolstering pathogenicity, and mediating interactions with the environment. Fungal EVs have emerged as promising candidates for innovative applications, not only in the management of mycoses but also as carriers for therapeutic molecules. Yet, numerous questions persist regarding fungal EVs, including their mechanisms of generation, release, cargo regulation, and discharge. This comprehensive review delves into the present state of knowledge regarding fungal EVs and provides fresh insights into the most recent hypotheses on the mechanisms driving their immunomodulatory properties. Furthermore, we explore the considerable potential of fungal EVs in the realms of medicine and biotechnology. In the foreseeable future, engineered fungal cells may serve as vehicles for tailoring cargo- and antigen-specific EVs, positioning them as invaluable biotechnological tools for diverse medical applications, such as vaccines and drug delivery.

## 1. Introduction

Extracellular vesicles (EVs) are defined as cup-sized nanostructures delimited by lipid bilayers [1]. The International Society for Extracellular Vesicles (ISEV) proposed classifying EVs based on their sedimentation speed [2], but most scientists and authors still use a former classification based on the origin and size of EVs, dividing them into exosomes and microvesicles (also called microparticles or ectosomes), which are shared across almost all domains of life [3], and apoptotic bodies, whose presence in fungi is currently under investigation [4]. The debate regarding EV classification is also still unresolved because there is not a universal consensus upon the most reliable methods of isolation, purification and characterisation [5,6,7,8,9,10]. The content of EVs can be very heterogeneous, and mainly includes proteins, lipids and nucleic material [11].

Since their early descriptions as “clotting factors” [12] or “platelet dust” [13], our knowledge on extracellular vesicles (EVs) has tremendously expanded, such that they are currently used in biotechnological, diagnostic and therapeutic applications [14,15,16]. However, several aspects of EVs, such as biogenesis, cargo and releasing mechanisms differ among species and are still far from being wholly understood [17]. Fungal EVs are a recent research field, since they were properly identified for the first time in 2007 in *Cryptococcus neoformans* [18]. Thanks to their cargo, EVs are used by fungal species to modulate a series of different functions within the fungal community [19]. For instance, they participate in intercellular communication during biofilm formation [20,21], regulate the intracellular proliferation of a population [22], play a critical role in cell wall remodelling [23] and could even mediate the vertical and horizontal transfer of prion-like protein in fungi [24]. In addition to all the intra-kingdom functions, fungal EVs are known to help releasing cells during their interactions with other organisms, such as other fungi and bacteria, or with the host, including animals and plants [4,25]. The interactions between fungi and their host through fungal EVs have been studied especially during infections in mammalian cells, where fungal EVs can exacerbate or attenuate fungal infection by enhancing pathogenicity or modulating virulence strategies [2,26,27,28]. Fungal EVs show the capacity to enter host cells and start a process that can lead to the modulation of antimicrobial activities and immune responses. Several studies investigated the effects of EVs from pathogenic fungi on murine and human immune cells, including macrophages [29,30], keratinocytes [31], dendritic cells (DCs) [32] and neutrophils [33]. These interactions lead to the activation of the innate immune system, but also of the priming activity of T cells through several mechanisms, including the production of pro-inflammatory and anti-inflammatory cytokines.

In addition to fungi, the bigger portion of the literature about the interactions between microbial EVs and the host immune system relies on bacteria. In the last twenty years, there has been a great number of studies regarding EVs both from pathogens [34] and commensal bacteria [35]. Several Gram-negative commensals have been investigated [36,37], while Gram-positive studies have focused on probiotics of the genera *Lactobacillus* and *Bifidobacterium* [38,39]. Research on EVs in Gram-positive bacteria, mycobacteria and fungi was neglected until recently, due the presumption that vesicles could not traverse the thick cell walls found in these organisms [40]. Despite the later discovery, EVs from these microorganisms are now comprehensively studied, and share common features, including the delivery of multiple virulence factors that elicit strong immune responses in the host [41].

The current available information on the applications of EVs in drug delivery provides significant groundwork for their development as delivery vehicles to harness their potential for future therapeutic applications. Fungal EVs could represent novel biotechnological tools for diagnostics and immunotherapy. Fungal EVs have been shown to possess most of the characteristics for an ideal vaccine, such as nanometric sizes, being g naturally carriers of multiple antigens, and conjugation with sugars, which can contribute to cell recognition [42].

The increased number of immunocompromised patients [43], and the spread of new fungal pathogens due to environmental and climate change [44,45,46] have significantly modified the rate of fungal infections among humans around the globe, as attested by the Global Action Fund for Fungal Infections (GAFFI) [47,48]. These dramatic data, together with the low efficacy of currently approved antifungal drugs and the outbreak of resistant fungal strains [49,50,51], outline the urgent need for new strategies to both prevent and fight fungal infections.

Despite the promising results and the comprehensive studies on EVs produced by both yeast-like and filamentous fungal species, this field of study is still in its infancy. Several questions are still to be understood.

The present review will describe the current state of knowledge, and recent insights into the molecular, chemical and immunological properties of fungal extracellular vesicles, focusing on the potential use of fungal EVs in antifungal diagnostics, vaccines and therapy.

## 2. Fungal EVs: Discovery, Composition and Release

Fungal EVs have been described in more than twenty different species [19], and it is now assumed that they represent a universal mechanism for the transport of molecules outside the fungal cell [52,53]. Nevertheless, we still lack data on their properties, biogenesis and functions, and multiple comprehensive reviews have focused on these open questions [4,19,40,54,55,56,57,58]. A brief history of their discovery, along with the current knowledge about the mechanisms of their production and cargo loading will be discussed in this chapter.

### 2.1. Discovery and Seminal Studies

The biological entities that would later be named extracellular vesicles were first described in 1946 by Chargaff and West, who were working on blood coagulation [12]. Only in 1967 did another study by Wolf describe them as platelet-derived material [13]. In fungi, the term “extracellular vesicles” was used for the first time in 1977, in a study on *Candida tropicalis* [59]. Before that, both Gibson and Peberdy in 1972 [60] and Takeo and colleagues in 1973 [61] reported the presence of outer membranous particles in *Aspergillus nidulans* and *Cryptococcus neoformans*, respectively. Despite the fact that in 1990 Anderson and colleagues found wall-crossing vesicles in *Candida albicans* [62], all of the aforementioned studies did not promote research on fungal EVs, mostly because the thick cell wall of fungi seemed to preclude the production of vesicles destined to the extracellular space [40]. The first set of experimental evidence suggesting the production of EVs by fungi was carried out at the end of the last century, when an attempt to develop monoclonal antibodies against *Cryptococcus neoformans* was made by Arturo Casadevall, as reported in the footnote of the first book on fungal EVs [63]. Since lipid structures were found in the fungal cell wall, Rodrigues and Nimrichter hypothesised the presence of hydrophobic structures that could reach the cell wall [64]. Following this hypothesis, in 2007 Rodrigues and colleagues isolated EVs from *C. neoformans* cultures [18], and right after this they demonstrated that these vesicles contain glucuronoxylomannan (GXM), which is the major virulence factor of *C. neoformans* [65]. Around the same period, Vallejo and colleagues showed that EV production occurs in *Paracoccidioides brasiliensis* [66]. To date, EVs have been found in more than 20 fungal species, in both yeast-like and filamentous forms, such that the production of EVs is now assumed to be a communication mechanism shared by the whole fungal kingdom. An almost complete list of these species can be found in previous comprehensive reviews [4,19], except (as far as we know) for the pathogenic yeast *Candida auris*, whose EVs have been characterised in the last year [67,68,69]. Fungal EVs have been investigated mostly in human infections and, more generally, in the context of interactions with the human host; therefore, this field of study will be deepened in a following chapter of this review. Nonetheless, fungal EVs have been studied also regarding their functions in parasitic or mutualistic relationships with plants [25,70,71], defence from predators [72], cell wall synthesis [73] and remodelling [23,66,74], and other physiological roles within fungal communities [75], including morphological transitions during biofilm formation [20,21], the regulation of protein cargo related to nutrients [76,77,78,79] and adaptation to the matrix [80,81,82].

### 2.2. Biogenesis and Release of Fungal EVs

Despite the fact that studies on the production of fungal EVs have been conducted for more than a decade, the mechanisms of their biogenesis are still unknown. Nevertheless, proteomic, genetic and in silico analyses have provided insights into certain molecular patterns related to EV trafficking, which are already extensively discussed elsewhere [4,19,27,57,83,84,85].

Mammalian vesicles, which have been far more frequently investigated than fungal ones have, are finely classified in different subpopulations, depending on the size and origin [86,87,88,89]. Currently, studies on fungal EVs also divide them into exosomes, which correspond roughly to small EVs in the ISEV classification and are smaller than 150 nm, and microvesicles, which correspond to medium EVs and are larger than 150 nm [27]. The most important difference between exosomes and microvesicles, apart from size, is the mechanism of biogenesis. Exosomes derive from the endosomal pathway, through the formation of multivesicular bodies (MVBs) which contain several vesicles. They fuse with the plasma membrane releasing them into the intraluminal space. On the other hand, microvesicles (or ectosomes) originate directly from projections of the plasma membrane [57].

Exosome biogenesis, release and cargo selection have been shown to be related with both conventional and unconventional secretory pathways, thanks to genetic deletions, but none of the studied mutations were able to completely cease EV production, suggesting a co-occurrence of the multiple known pathways or the presence of currently unknown ones. Firstly, mutations in SEC genes, which play a role in the post-Golgi conventional secretion pathway, cause an accumulation of vesicles in the cytoplasm [90,91] or vesicles of altered diameters [92] in *S. cerevisiae*, and a lack of EV detection in *C. neoformans* [93]. Regarding the unconventional secretion pathways, the most important regulators of EVs production known so far are involved in MVB formation and release. Specifically, a lack of expression of Golgi reassembly and stacking protein (GRASP) causes a reduction in EV release in *S. cerevisiae* [92] and different EV sizes in *C. neoformans* [94]. Moreover, a class of proteins of vacuolar protein sorting (VPS) named E-VPS is crucial in the endosomal sorting complexes required for transport (ESCRT) machinery [95,96,97], which coordinate MVB release [98]. Therefore, it has been shown that the deletion of different VPS genes strongly affects the biogenesis of EVs in *S. cerevisiae* [23,92], *C. neoformans* [99,100,101,102] and *C. albicans* [21,103].

The production and release of microvesicles via the invagination of the plasma membrane was shown two decades ago [64], together with observations of cytoplasmic volume loss [92] and cytoplasmic proteins in fungal EVs [104]. Moreover, mutations in genes involved in the composition of the plasma membrane affect EV formation and content [81,105,106,107]. Nevertheless, we are still largely unaware of the molecular process for the formation of fungal microvesicles at the plasma membrane level.

Another fungal EV-related process that still needs to be fully understood is cell wall crossing. The fungal cell wall is composed of a core set of molecules that are conserved across the fungal kingdom, mainly beta-glucans, mannoproteins and chitin [108], and of components that are species-specific, such that the wall is probably the part of the cell that exhibits the most diversity and plasticity [109]. Fungal cell walls are nowadays considered dynamic structures whose compositions and pores’ dimensions are highly regulated via environmental conditions [55,110]. Despite this, they have been shown to act as barriers to the passage of EVs, since mutations in cell wall-remodelling genes increase their release [23,102], and an increase in their thickness can lead to an accumulation of vesicles in the area between the plasma membrane and cell wall [111]. Moreover, it still remains poorly understood how vesicles as large as almost 1 μm could pass through these dense polysaccharide/protein matrices. Three nonexclusive hypotheses, already reviewed elsewhere [40,55], are currently under investigation: (i) vesicles can pass the cell wall through guide channels; (ii) there are specific enzymes that could open areas in the wall; (iii) turgor pressure and/or other physical forces could force the passage of EVs through cell wall pores. The last hypothesis has received support via a recent study which demonstrated that the cell wall has viscoelastic properties, which could allow the passage of EVs [112]. However, since that study utilised small liposomes (60–80 nm), the full explanation for the passage of EVs with a broad spectrum of sizes has yet to be found.

### 2.3. Composition of Fungal EVs

The process of the production and release of EVs is conserved across kingdoms even if it has a considerable energetic cost for the cell, and the reason for that has to be found in the multiple advantages of vesicle packaging compared to soluble secretion directly to the extracellular milieu. Firstly, vesicles protect easily degradable molecules, such as RNAs. Secondly, loading molecules into a confined space prevents the loss of them. Finally, the vesicle co-transportation of more molecules enables their synergistic effect over distances [113]. As stated before, a large part of the knowledge on fungal EVs is derived from studies on mammals, since there are several resemblances between them, including their content in terms of classes of molecules [55]. The most abundant molecules packed the into EVs of both mammals and fungi belong to proteins, lipids and nucleic acids (shown in Figure 1), even though there are other molecules specific to fungi, such as prions. Most of them have been summarised in two recent and comprehensive reviews [4,114], and here we review them briefly.

#### 2.3.1. Proteins

Proteomic studies on fungal EVs showed a common set of proteins between different species, such as those involved in the stress response, oxidation, metabolic pathways, transport and signalling, while others have proven to be species-specific [53,113,119].

EVs from pathogenic species carry proteins associated with virulence. In *C. neoformans*, laccases, ureases, phosphatases and heat shock proteins were found [65]; in *Paracoccidioides brasiliensis*, phosphatases [119]; in *Histoplasma capsulatum*, catalases, superoxide dismutases and cell-wall hydrolyzing enzymes [74]; in *Malassezia sympodialis*, multiple allergens [122]; and in *C. albicans*, numerous proteins related mainly to pathogenesis, cell organisation and the response to stress [123,124].

The protein content of non-pathogenic yeast EVs have also been investigated, especially in *S. cerevisiae*, *Torulaspora delbrueckii*, *Hanseniaspora uvarum*, *Candida sake* and *Metschnikowia pulcherrima* [92,120]. Researchers have found proteins implicated in almost every cell pathway, but the most present were heat shock and stress-related proteins, and those involved in cell wall metabolism [121]. Moreover, proteins contained in EVs from Pichia fermentans were shown to be related to dimorphic transition [20].

#### 2.3.2. Lipids

Lipids are the major components of vesicles’ membranes, and also part of their cargo. Several lipids found in fungal EVs have important functions in pathogenesis, biofilm building and membrane formation [113,126,131]. For example, ergosterol, which is commonly found in EVs from all fungal species, is known to be crucial for biofilm building [131]. EVs from *C. albicans* and *C. neoformans* are enriched in ergosterol and glucosylceramide [106,124,127], an important immunogenic compound which has been proven to be a good target for inhibiting fungal hyphal formation during pathogenesis [64]. EVs from *H. capsulatum* and *P. brasiliensis* were found to mainly contain sterols and phospholipids, such as phosphatidylethanolamine, phosphatidylcholine, and phosphatidylserine, which are key elements in the formation of fungal lipid bilayers [74,126,128].

#### 2.3.3. Nucleic Acids

Concerning nucleic acids, fungal EVs show strong resemblance to those of mammals, transporting several types of both coding and non-coding RNA, which have been compared and reviewed elsewhere [4,132,133,134]. RNAs are receiving special attention in the context of intercellular communication due to the several mechanisms of post-transcriptional regulation, such as those mediated by microRNAs (miRNAs), short interfering RNAs (siRNAs), tRNA-derived fragments (tRFs) and long non-coding RNAs (lncRNAs) [135,136]. However, they are commonly found in EVs across all biological kingdoms [25,137,138,139]. For instance, several studies showed cross-kingdom interactions between fungi and plants through RNA-containing EVs. Pathogen-induced gene silencing (PIGS) is a mechanism of RNAi used by several fungal pathogens [136], such as *Botrytis cinerea* [140], *Verticillium dahliae* [141], *Puccinia striiformis* f.sp. *tritici* [142] and Fusarium graminearum [143].

Apart from studies on plants, several studies were conducted on yeast species related to humans. The first one was a comparison between the RNA content of five different species, as reported by Da Silva and colleagues in 2015 [115]. These authors found that the most common shared small RNA classes in EVs belong to small nucleolar RNAs (snoRNAs), small nuclear RNA (snRNAs) and tRNA-derived fragments (tRFs). Then, EVs from *Malassezia sympodialis* were shown to contain small RNAs similar to miRNAs [116], which was also the case in Pichia fermentans [20], while *Histoplasma capsulatum* EVs contained 25 nt long anti-sense RNAs [117]. Moreover, a few studies have indicated that the EV-RNA composition differs from the cellular composition [94,144]. The RNA content of EVs from *P. brasiliensis* and *P. lutzii* shared mRNA sequences related to protein modification and DNA metabolism, as well as small non-coding RNAs, some of which could potentially modulate the murine immune response [118]. A study on *Cryptococcus gattii* focused on the impact of the membrane architecture on the formation and content of EVs revealed that the RNA cargo was strongly altered in mutant cells [81]. Finally, a recent study comparing the EV cargos and functions of *Candida albicans*, a human-related yeast, and *Candida auris*, a new fungal pathogen that has emerged as a result of global warming [45], found significant differences in RNA content and speculated that these differences could explain the phenotypic changes induced by these EVs in human cells during immunological assays [67].

#### 2.3.4. Other Molecules

Other than proteins, lipids and nucleic acids, the most abundant molecules found in fungal EVs are carbohydrates, pigments and prions. Glucuronoxylomannan (GXM) has been found in EVs from *C. neoformans* [18], as described before in this review, and in *C. gattii* [81,129]. Moreover, other polysaccharides have been found in EVs from *P. brasiliensis* [66], *P. lutzii* [130] and *C. albicans* [21]. *C. neoformans* EVs were also found to carry melanin in an older study [145]. Prions have been found in EVs from *S. cerevisiae*, both in soluble and aggregated forms [24,125], and it has been hypothesised that they can play a role in vertical and horizontal transmission [146].

## 3. Comparisons between Bacterial, Mycobacterial and Fungal EVs

The discovery of EVs in prokaryotes reinforces the hypothesis that vesicular transport is an universal mechanism. Bacterial EVs were documented for the first time in *Escherichia coli* in the 1960s [147,148,149,150], but research in the field of EVs from Gram-negative bacteria has increased substantially in the last twenty years. On the contrary, EVs produced by Gram-positive bacteria were described for the first time in 1990 [151]. The delay in their discovery was due to the same unverified inference regarding fungal EVs, i.e., that the thickness of these organisms’ cell walls should not allow the release of vesicles.

EVs from Gram-negative bacteria are derived from the outer membrane and are thus generally considered outer-membrane vesicles (OMVs) [152]. OMVs are released via a pinching-off process of the outer membrane, encapsulating components from the periplasmic space [153,154,155,156,157]. OMVs may be produced during cell wall turnover, as a result of increased turgor pressure from cell wall components, such as peptidoglycan, or when repulsion between charged B-band lipopolysaccharide (LPS) molecules induces membrane budding [158,159,160]. In *Pseudomonas aeruginosa*, OMV formation may be induced via the mechanisms of quorum-sensing molecules (e.g., the *Pseudomonas* quinolone signal) that favour the formation of membrane blebbing [161,162].

It was observed that OMVs can play a major role in microbial pathogenesis. In general, the OMV cargo includes virulence factors, adhesins, DNA, RNA, communication compounds, toxins, immunomodulatory factors and nutrient-scavenging factors. For instance, the OMVs of *E. coli* and *P. aeruginosa* carried in the Shiga toxin and Cif toxin, respectively [163,164]. These vesicles were found in association with cytotoxicity, the invasion of host cells, membrane fusion, the production of biofilms, and the transfer of viruses, DNA, receptors and antibiotic resistance proteins [153,165,166,167]. Interestingly, many of these different functions of OMVs are shared with the EVs produced by Gram-positive bacteria, mycobacteria and fungi.

In 2009, the protein composition of EVs produced by the Gram-positive *Staphylococcus aureus* was characterised via mass spectrometry, showing, for the first time, that the size of EVs from a Gram-positive microorganism ranged from 20 to 100 nm in diameter. This size was comparable to that of EVs isolated from Gram-negative bacteria. EVs from *Bacillus* spp., *Clostridium perfringens* and *Streptomyces. coelicolor* range from 20 nm to 400 nm in diameter [168,169,170,171]. These findings suggest that, although vesiculogenesis may be a universal phenomenon, different microorganisms synthesise and regulate EVs in different ways.

While there is no physical barrier to the release of OMVs from Gram-negative bacteria to the extracellular space, Gram-positive bacteria lack an outer membrane but have a thick peptidoglycan cell wall outside of the cell membrane [172]. Mycobacteria also release EVs as a means to secrete complex groups of proteins and lipids into the extracellular milieu [173,174,175,176]. Vesicle-like blebs were observed on the surface of mycobacterial cells via TEM and SEM and were similar in size to that of purified EVs [177]. In mycobacteria, the cell wall is composed of peptidoglycan covalently attached to arabinogalactan, which in turn binds mycolic acids. Free lipids are associated with the upper segment of this cell wall that is surrounded by an outermost capsule composed of polysaccharides, proteins and lipids [178]. Some studies focused on EV production in the more medically important strains of mycobacteria, such as *Mycobacterium tuberculosis* and *Mycobacterium bovis* bacille Calmette–Guérin (BCG), showing that the size distribution of EVs isolated from *M. bovis* BCG, ranging from 50 nm to 300 nm in diameter, was similar to that of OMVs from Gram-negative bacteria and that of EVs from most Gram-positive bacteria [179]. Although a recent study of mycobacterial EV production under iron-limiting conditions indirectly supports the role of EVs in virulence [180], other mycobacterial strains, including non-pathogenic and fast-growing strains, also produced EVs.

The structure of Gram-positive bacteria and mycobacteria is analogous to that of fungi. The fungal cell wall consists of semi-striated layers of chitin, β-glucans and mannoproteins [181]. Some fungi also contain melanin in their walls, although it is unknown whether this molecule is permanent or transitory [18]. The pore size of the *S. cerevisiae* cell wall varies by strain from 50 to 200 nm, but can change based on the cell wall remodelling enzymes, extracellular pH, culture growth phase, and accumulation of deposits such as melanin under distinct conditions. As already mentioned, the presence of these thick cell walls hindered the search for EVs.

Three nonexclusive hypotheses have been formulated to explain the enigma of how vesicles cross the fungal cell wall: (i) channels in the cell wall allow the EVs to cross this barrier moving towards the extracellular environment, (ii) cell wall-remodelling enzymes temporarily modify the cell wall to allow release of EVs, and (iii) uncharacterized mechanical or other physical forces push vesicles through small pores.

The identification of virulence factors in the EV cargo suggest a role of these membranous structures in fungal pathogenesis. However, in the non-pathogenic yeast *Saccharomyces cerevisiae*, EVs were identified [92]. To date, EVs have been described in many fungi as previously described in this review, and proteomic studies have indicated over 400 protein cargos, including proteins that have a role in cell metabolism, signal transduction and virulence, as well as structural scaffold proteins and nuclear proteins. Therefore, the role of microbially released EVs in eukaryotic microbes is presumably similar to that of those in bacteria.

## 4. Fungal EVs and Interactions with Host Immunity

Extracellular vesicles are well-known mediators of the immune response [86,182,183] thanks to their immunogenic cargo. Several studies, previously reviewed [4,19,26,184,185,186], explored the immunomodulation potential of EVs from both pathogenic and non-pathogenic fungal species. It is now well established that fungal EV cargos are recognised as pathogen-associated molecular patterns (PAMPs) by the pattern recognition receptors (PRRs) of the host innate immune system, and that therefore fungal EVs could modulate the activation of host immunity [187]. Taken together, the studies on the immunomodulatory properties of fungal EVs show that they play an important role in host–fungal communication, both during infection and commensalism dynamics. Fungal EVs are recognised by the host cell and activate an immune response that may depend on both the fungal species and the EV composition [185]. Here, we present the current data on virulence factors contained in fungal EVs, as well as their immunomodulation properties. A summarising picture is shown in Figure 2. Altogether, these data support the investigation of fungal EVs for their potential to be used as vaccines against fungal infections. The literature on this topic will be reviewed in chapter 6 of the present review.

### 4.1. Promoting Infections

The induction of infection by fungal pathogens is mediated by virulence factors, which promote crucial mechanisms such as adhesion, invasion, host damage and immune evasion [195,196,197]. Many of these virulence factors, packed into EVs such that they were called “virulence bags” in early studies [53,65], are delivered to target host cells. Here, we provide an overview of the virulence factors often observed to be associated with pathogenic fungal EVs and their mechanism of action, both of which have been exhaustively reviewed elsewhere [185].

*Candida albicans* secretes aspartic proteinases (Sap) and agglutinin-like sequence proteins (Als), which are both important virulence factors [194]. Sap proteinases mediate penetration, nutrient acquisition and immune evasion [198,199], while Als proteins are important for adherence, biofilm formation and dimorphic transition [200,201].

Several human pathogenic species belonging to the genus *Cryptococcus*, such as *C. neoformans* and *C. gattii*, are known for the production of a polysaccharide capsule and melanin, which are both recognised as virulence factors [192]. GXM, the principal component of the capsule of *C. neoformans* [202], exerts immunosuppressive effects on multiple leukocytes and promotes the death pathway in macrophages [203,204,205], while melanin confers protection to fungus-neutralising reactive oxygen species (ROS) [206].

Moreover, EVs from highly virulent strains of *C. neoformans* can enhance the pathogenicity of less virulent strains through the delivery of virulence factors [193]. The same results have been observed in *C. gattii*, a species that was already shown to present a mechanism called “division of labour”, which consists of the induction of the rapid intracellular proliferation of non-virulent strains by a highly virulent strain in a co-infection model [22]. A later study proved that EVs from the highly virulent strain of *C. gattii* are major players in virulence, enhancing the less virulent strain infecting macrophages. This mechanism was exerted through EVs entering the immune cell and raising the intracellular proliferation rate (IPR) of the fungus without an augmentation of phagocytosis [129].

Heat shock protein 60 (HSP60) and catalase B are commonly found in the EVs of *Histoplasma capsulatum*. These virulence factors are essential for the survival of the fungus during infection, since HSP60 interacts with complement receptor 3 (CR3) of human macrophages and mediates CR3-based phagocytosis, a mechanism not sufficient to induce oxidative burst [197,207,208]. This strategy allows the fungus to survive and replicate inside phagosomes, also thanks to the activity of catalase B, which inhibits ROS [209]. Infections of *Paracoccidioides brasiliensis* are enhanced by glycoprotein 43 (gp43) [210], whose functions are the promotion of fungal adhesion both to the extracellular matrix and to the cell [211,212], and the contrast of fungal killing through the inhibition of phagocytosis and nitric oxide (NO) production [213].

In addition to being virulence factors, fungal EVs have proven to directly facilitate pathogenesis in three other studies so far. EVs from *C. neoformans* enhanced fungal passage through the blood–brain barrier (BBB), thanks to a proposed mechanism of fusion between EVs and lipid rafts from brain microvascular endothelial cells [214]. Moreover, treatment with EVs from a capsular strain of *C. neoformans* after infection with the same strain in mice strongly enhanced the fungal burden after two weeks, despite an initial reduction [28]. The same treatment in larvae of *Galleria mellonella* led to uncontrolled infection [215]. Similar results were obtained with EVs from *Sporothrix brasiliensis*, whose inoculation during an infection from the same species produced a higher fungal load and larger lesions [32].

### 4.2. Modulating Immunity

Promoting infection is not the only possible consequence of the interaction between fungal EVs and the host. The uptake of EVs’ rich cargo by the immune cells of the host could lead to multiple effects, including a containment of the infection or tolerance of the fungus by the immune system. Here, we review the most important findings regarding the immunomodulatory mechanisms and effects of fungal EVs, originating from different fungal genera and species.

#### 4.2.1. Cryptococcus

Fungal EVs of *C. neoformans* were the first ones investigated in 2008 [65], and since then several studies have focused on the functional characterisation of EVs from this species. In addition to the aforementioned works, a study in 2010 by Oliveira and colleagues [29] showed multiple effects of EVs from either capsular or acapsular strains of *C. neoformans* on murine macrophages. After internalisation, EVs from capsular strains induced the production of anti-inflammatory cytokines, such as transforming growth factor β (TGF-β) and IL-10, therefore impairing the production of IL-12 and the intracellular killing of the fungus, which is a known immunosuppressive effect of the capsule component GXM [203,216]. Instead, EVs from acapsular strains (i.e., without GXM) induced a greater production of NO and TNF-α, as well as an increased ability to phagocytize and kill fungal cells. Thus, this study showed that the diverse cargo in EVs from *C. neoformans* induce both the positive and negative stimulation of macrophages. Moreover, the presence of specific antibodies against proteins packed in EVs from *C. neoformans* in the sera of patients diagnosed with cryptococcosis demonstrated the activation of the humoral immune response against these EVs [65].

#### 4.2.2. Candida

In addition to EVs from *C. neoformans*, several studies have been conducted on EVs from *C. albicans*, which is the most common fungus in the human microbiota [217], and is able to establish both commensal and pathogenic relationships with the host, as consequences of several factors [218,219]. These studies show that EVs from *C. albicans* enhance the activation of the innate immune response. In a study by Vargas and colleagues in 2015 [124], murine RAW 264.7 macrophages stimulated with EVs produced high amounts of IL-12p40 and NO, and lower concentrations of IL-10 and TGF-β, while bone marrow-derived macrophages (BMDM) produced NO, IL-12p40, TNF-α and IL-10. In the same work, bone marrow-derived dendritic cells (BMDC) stimulated with EVs produced IL-6, IL-10, IL-12p40, TNF-α, and TGF-β, a result also confirmed by Zamith-Miranda and colleagues in 2021 [67]. BMDCs underwent cell maturation, increasing the expression levels of some cell surface markers of activation, such as major histocompatibility complex class II (MHC-II) and CD86. These findings suggest that the EVs could also elicit the immune adaptive response against the pathogen.

The ability of EVs from *C. albicans* to activate adaptive immunity was demonstrated by Vargas and colleagues in 2020 [191]. After an initial treatment with *C. albicans* EVs, subsequent infection of mice with the fungus was reduced such that the treated mice survived candidiasis, whereas the control subjects died. The inoculation of EVs produced an increase in immunoglobulin G (IgG) levels against the protein packed in EVs, as already shown both in mice and humans [123,124] accompanied by an increased production of IL-12p70 and TNF-α, despite a surprising increase of anti-inflammatory cytokines such as IL-4, IL-10 and TGF-β also. It has been hypothesised that the phospholipids in *C. albicans* EVs could be responsible for immunity activation, since deleting the phosphatidylserine synthase gene resulted in EVs unable to activate the nuclear factor kappa-light-chain-enhancer of activated B cells (NF-κB) in macrophages [106]. EVs from *C. auris* were also able to stimulate IL-6 secretion from BMDCs and reduce the production of TGF-β [67].

#### 4.2.3. *Histoplasma capsulatum*

Immune modulation by EVs from *H. capsulatum* was investigated mainly in the context of a pre-treatment of the fungus with monoclonal antibodies (mAb). An early study showed that HSP60, a protein packed into EVs from *H. capsulatum*, was recognised by antibodies from the sera of patients with histoplasmosis [74]. Moreover, it was shown that pre-treatment with mAbs against HSP60 significantly prolonged the survival of mice infected with *H. capsulatum* [220]. Moreover, Baltazar and colleagues observed that EVs produced after the pre-treatment of *H. capsulatum* with two mAbs against HSP60 were larger and had more abundant total protein cargo and significantly reduced phosphatase and laccase activity [128]. Finally, EVs from treated fungal strains reduced phagocytic rates and intracellular fungal killing by BMDMs [188]. Thus, since it is now clear that the HSP60 protein loaded into *H. capsulatum* EVs is a major player in the survival rate of the pathogen in the host, it could represent one of the most important targets for antibody-based therapy against histoplasmosis.

#### 4.2.4. *Malassezia*

The genus *Malassezia* includes dimorphic species that commonly colonise human skin both as a commensal and pathogen of skin-related diseases, such as dermatitis and atopic eczema [221]. EVs from *M. furfur* were shown to penetrate human keratinocytes in vitro and the damaged skin of mice in vivo, producing an upregulation of IL-1β and IL-6 in both the models [222]. The stimulation of human peripheral blood mononuclear cells (PBMCs) from healthy individuals with EVs from *M. sympodialis* resulted in increased levels of TNF-α, while in patients with atopic eczema this resulted in an increased production of both IL-4 and TNF-α [190]. Interestingly, DCs were shown to be able to phagocytize the EVs and produce their own EVs containing *M. sympodialis* antigens. In the same study, immunoglobulin E (IgE) from patients with atopic eczema was shown to react with antigens present in EVs from *M. sympodialis*. The presence of IgE-stimulating antigens in EVs from this species was also confirmed via a study by Johansson and colleagues [122]. They showed that EVs were also internalised by human keratinocytes and monocytes. Moreover, keratinocytes from healthy individuals treated with EVs from *M. sympodialis* increased ICAM-1 production [223], which mediates both T cell and neutrophil migration to the site of infection [224,225].

#### 4.2.5. *Paracoccidioides*

Some species of the genus *Paracoccidioides*, namely *P. brasiliensis* and *P. lutzii*, are dimorphic fungi that could cause disease in humans, specifically Paracoccidioidomycosis [226,227]. EVs from *P. brasiliensis* have been demonstrated to induce M1 polarisation in macrophages, and even to revert a M2 phenotype toward a M1 one [30]. This modification plays a protective role in paracoccidioidomycosis progression, being responsible for fungal clearance and the production of inflammatory cytokines, such as NO, TNF-α, IL-6, IL-12p40, IL-12p70, IL-1α, and IL-1β [228]. Moreover, an early study tested antigens of EVs from *P. brasiliensis* with sera from paracoccidioidomycosis patients, resulting in the generation of specific antibodies against these antigens [66].

#### 4.2.6. Aspergillus

*A. fumigatus* and *A. flavus* are the most common causative agents of aspergillosis [229,230]. Two significant studies were conducted to test EVs; immunomodulation activity on human phagocytes. *A. fumigatus* EVs induced an increased production of proinflammatory cytokines in macrophages in vitro, and thew pre-treatment of both macrophages and bone marrow-derived neutrophils (BMDN) with the same EVs produced a higher phagocytic index and caused higher intracellular killing in an infection model with the fungus [189]. Interestingly, low amounts of EVs protected the fungus against intracellular killing. A study by Brauer and colleagues [33] obtained similar results with *A. flavus* EVs. BMDMs stimulated with those EVs secreted high amounts of NO, TNF-α, IL-6, and IL-1β, following classical M1 polarisation.

#### 4.2.7. *Sporothrix*

A study by Ikeda and collaborators showed that EVs from *Sporothrix* spp. displayed few immunomodulatory effects on BMDCs. However, treatment with EVs from *S. brasiliensis* and subsequent exposition to the fungus resulted in IL-12p40, TNF-α, and IFN-γ production, as well as an increased phagocytic index followed by a rise in fungal burden within the cells [32]. The same study showed that the in vivo treatment of mice with *S. brasiliensis* EVs, before infection with the fungus, produced larger lesions and specific antibodies against those EVs compared to those of un-treated mice.

#### 4.2.8. *Trichophyton*

Species of the genus *Trichophyton* are associated with superficial mycoses in human [231]. EVs from *T. interdigitale* induced M1 polarisation in macrophages, as well as they increased phagocytosis and the killing of the fungus in subsequent infection. At the same time, both macrophages and keratinocytes treated with these EVs displayed a higher production of proinflammatory cytokines [31].

## 5. Fungal EVs for Therapeutical Applications

In the previous chapter, we described the currently available literature on the interactions between EVs from diverse fungal species and the immune system of humans and mice. Despite the growing body of work on this topic in the last fifteen years, many questions remain unanswered. For instance, the mechanisms underlying the diversity of molecules within fungal EVs, which lead to a large spectrum of immune responses, is still a matter of debate. In fact, depending on the fungal species and environmental conditions, some fungal EVs could be beneficial to the host, whereas others could enhance the development of disease. Nonetheless, the existing studies have already provided enough clues with which to consider the properties of fungal EVs as suitable for their use in therapy, such that it has been proposed to explore fungal EVs as versatile tools in different therapeutic fields, especially as drug targets and in chemotherapy or vaccine development [26]. To date, there are few studies dissecting fungal EV applications in these proposed fields, but as recently reviewed, their physical, chemical and biological properties give them enormous potential to be exploited for human health [42]. Moreover, given the global number of deaths due to invasive mycoses, which have also been caused by the emergence of new fungal pathogens and resistance to antifungal drugs (around 1.5 million per year), and the fact that the only two existing antifungal vaccines are still under development, there is a great urgency to build strategies that could help in both preventing and curing fungal infections [49,232,233]. Here, we review the most prominent data on fungal EVs both as potential biomarkers and players in vaccine development.

EVs produced by humans are under development for being used as biomarkers for several conditions, especially for multiple types of cancer [234,235]. Furthermore, fungal EVs are beginning to receive attention from the scientific community for this application. As shown before, EVs from several fungal species are recognised by the human immune system for their ability to interact with host cells through delivering their cargo and eliciting host immune reactions. As stated in the previous paragraphs, EVs of pathogenic fungi such as *C. neoformans*, *H. capsulatum*, and *P. brasiliensis* were proven to be sensed in human serum from patients with fungal infections (including cryptococcosis [65] histoplasmosis [74] and paracoccidioidomycosis [66], but not from healthy individuals. These findings offer a promise to be helpful in the context of fungal diagnostics, where the current limitations of traditional methods for the detection of invasive fungal infections [236,237,238], such as low sensitivity in early stages or missed diagnoses using blood cultures [239,240], often lead to considerable delays in the initiation of targeted treatment [241]. Some fungal antigens, such as galactomannan and β-glucan, have been exploited as markers for fungal infection diagnosis to overcome the aforementioned limitations [242]. Given the rich antigenic cargo of fungal EVs, a reliable EV-based diagnostic method could be a new and crucial tool in the relatively young field of fungal diagnostics.

Despite the availability of several antifungal drugs, such as azoles, pyrimidines, echinocandins and polyenes [243], the current number of deaths for fungal diseases, which is already high, is expected to significantly grow due to the increase in drug resistance [49]. Moreover, the currently used antifungal drugs are expensive, not available in every country, and have a treatment time that takes several months [48]. Therefore, a crucial field the near future is represented by the discovery of new drugs, as well as the enhancement of antifungal vaccine research. The use of vaccines for preventing fungal infection has been less frequently exploited compared to that for other pathogens, such as bacteria and viruses [244], but in recent years some progress has been made, especially focusing on the great number of different fungal antigens [232]. In this context, several studies showed that fungal EVs have most of the needed features to be used for vaccine development. Bachmann and Jennings [245] identified two peculiar characteristics: (i) their role as cell-free systems, that are non-replicative and harmless for immunocompromised patients and (ii) their possession of multiple PAMPs for eliciting a sufficient immune response. The first studies that specifically addressed fungal EV potential in inducing an immune response against a fungus were conducted on the insect *G. mellonella*, which is a common animal model for antifungal molecules [246,247]. This species displays an innate response enhanced by a group of blood cells, called haemocytes, that are able to internalise the pathogens, and resemble phagocytic activity [248]. The vaccine potential of fungal EVs in this insect was tested in 2015 with EVs from *C. albicans* [124], in 2019 with EVs of *C. neoformans* [215] and in 2020 with EVs of *Aspergillus flavus* [33]. *G. mellonella* larvae were pre-treated with fungal EVs and then challenged with the fungal species. *C. albicans* and *A. flavus* EVs induced an improvement in survival rates and reduced fungal burden, whereas *C. neoformans* EVs only caused a delay in lethality. Such promising results led Vargas and colleagues to investigate the use of *C. albicans* EVs as a vaccine candidate against candidiasis in a model of immunocompromised mice [191]. After a treatment with three boosts of *C. albicans* EVs either alone or in combination with Freund’s adjuvant, mice produced IgG antibodies against fungal antigens and were able to survive a subsequent lethal dose of the fungus. Total protection against the fungal infection was reached both with and without the adjuvant, but in the first case a higher inflammatory response in terms of the cytokine profile was noticed. Moreover, it was observed that the treatment also reduced the fungal burden in the spleen, kidney and liver. Noticeably, *C. albicans* EVs were proven to maintain their properties after being stored at −80 °C. In 2021, more insights into fungal EV properties in vaccine development were provided by Rizzo and colleagues [249]. By using a recently developed protocol for EV isolation and new microscopy technologies [81], the authors analysed the cargos and structures of three different *Cryptococcus* species, revealing that these species produce EVs both with or without a surface decoration made of fibrillar structures being anchored to the lipid bilayer. Whereas capsular strains produced EVs that revealed the presence of GXM, as shown via anti-GXM fluorescent antibodies, acapsular strains also produced decorated EVs. Therefore, the authors hypothesised that the fibrillar structure is not composed of GXM, and reinforced this hypothesis showing that *C. albicans* and *S. cerevisiae* (which have neither a capsule nor GXM) also produced decorated EVs. After performing a proteomic analysis of *Cryptococcus* EVs, the authors concluded that their surface could be covered by mannoproteins and tested this hypothesis through fluorescence microscopy, showing that at least 95% of EVs were bound to mannoprotein-specific antibodies, whether they were from capsular or acapsular strains. Previous studies reported that the presence of mannose, as well as a vesicular size between 10 and 100 nm, could significantly improve EV uptake by host DCs [124,130,250], reinforcing the promising use of fungal EVs for vaccine development. Proteomic analysis in the same study on *Cryptococcus* [249] revealed the presence of immunogenic proteins previously tested as vaccine candidates against cryptococcosis [251,252]. Therefore, the authors performed a pilot experiment to address the use of EVs for immunisation against the same disease. BALB/c mice were treated three times with EVs from both capsular and acapsular *C. neoformans* strains, showing the production of antibodies against vesicular proteins in both conditions. After being challenged with a *C. neoformans* wild-type strain, EV-treated mice showed a prolonged survival rate compared to that of untreated mice, giving the best results with the highest dose of EVs from the acapsular strain. Thus, these results suggest that the EV cargo is decisive in the vaccination outcome. This assumption seems to also be confirmed by a study on *H. capsulatum* [188], in which the authors showed that the binding of mAbs to the fungus modified the cargo of its EVs, resulting for instance in a reduced presence of laccase activity. Moreover, macrophages treated in vitro with EVs, derived from the strain previously opsonized by antibodies, showed a decrease in the inflammatory response when challenged with *H. capsulatum* cells. Notably, this is a rare case where EVs from pathogenic fungi downregulate the host immune system, whereas the majority of the fungal species studied so far are proven to produce immunogenic EVs.

## 6. Conclusions

In recent years, fungal extracellular vesicles (EVs) have taken centre stage as crucial interkingdom signalling microstructures. Discoveries of their immunomodulatory properties have opened exciting avenues for their exploration in various pathogenic models. However, there is a pressing need to deepen our understanding of the genetic, biochemical, and physical aspects of fungal EVs to elucidate the intricate mechanisms governing their production, composition, and diversity. This is essential for achieving a comprehensive grasp of their structure and biological significance.

For instance, the following remain to be addressed: (i) stress signals or environmental changes able to change the composition of EVs; (ii) the molecular mechanisms of EV biogenesis; (iii) the deciphering of the massive presence of metabolic enzymes in fungal EV that have been described for all EV-producing fungi so far; (iv) the presence of distinct classes of RNA in EVs as observed in *C. neoformans*, *C. albicans*, *P. brasiliensis*, *S. cerevisiae* [115] and presumably in other fungal EVs; (v) the immunomodulatory properties and the leukocyte surface receptors responsible for EV recognition and internalisation.

The emergence of new fungal pathogens and resistance to antifungal drugs are global health security threats. Fungal EVs could represent a new strategy to prevent and treat fungal diseases given their immunomodulatory properties, their ability to transfer bioactive components [1,87,253] and ability to surmount biological barriers, including the blood–brain barrier (BBB) [254]. Strategies that use EVs for the control of fungal infectious diseases include host-induced gene silencing through RNA carried in EVs or artificial vesicles delivering RNA with antifungal properties [25].

The use of EVs as disease biomarkers and for drug delivery, or when bioengineered to act as vehicles in therapeutics has also been proposed [14,234,255,256]. Similarly to bacterial EVs, fungal vesicles have also been proposed as biomarkers, adjuvants, and vaccine or immunotherapy agents [19,26,257,258,259]. Although the aforementioned findings support the potential of fungal EVs to be used as vaccines, some concerns emerged from a few studies, such as the increased overgrowth of fungal cells, as observed in mouse brains when *C. neoformans* and its EVs were administered [214], after the immunisation of mice with EVs from *S. brasiliensis* was followed by subcutaneous infection with this pathogen [32], or again, when EVs from a nonencapsulated strain of *C. neoformans* induced a higher inflammatory response in macrophages when compared to EVs produced by the encapsulated strain [29]. In light of these discoveries, there is a compelling case for further research to identify suitable fungal candidates for the production of extracellular vesicles (EVs) intended for vaccine development. Genetic engineering holds the promise of tailoring these candidates to enhance their immunogenic characteristics and remove EV cargos that might contribute to fungal burden and disease advancement.

## Figures and Tables

**Figure 1 biomolecules-13-01487-f001:**
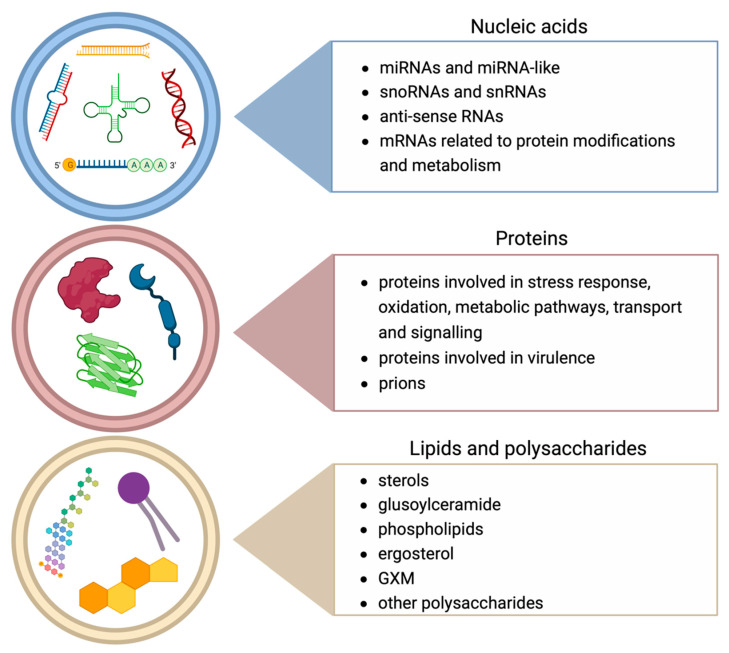
Most important classes of molecules carried by fungal EVs. The EV cargo is mainly composed of nucleic acids, proteins, lipids and polysaccharides. Nucleic acids found in EVs comprise miRNAs and miRNA-like [20,115,116], snoRNAs and snRNAs [115], anti-sense RNAs [117] and mRNAs related to protein modifications and metabolism [118]. Proteins comprise proteins involved in stress response, oxidation, metabolic pathways, transport and signalling [20,53,92,113,119,120,121], proteins involved in virulence [65,74,119,122,123,124] and prions [24,125]. Lipids and polysaccharides comprise sterols [74,126,127,128], glusoylceramide [106,124,128], phospholipids [74,106,124,126,128], ergosterol [106,124], GXM [18,81,129] and other polysaccharides [21,66,130].

**Figure 2 biomolecules-13-01487-f002:**
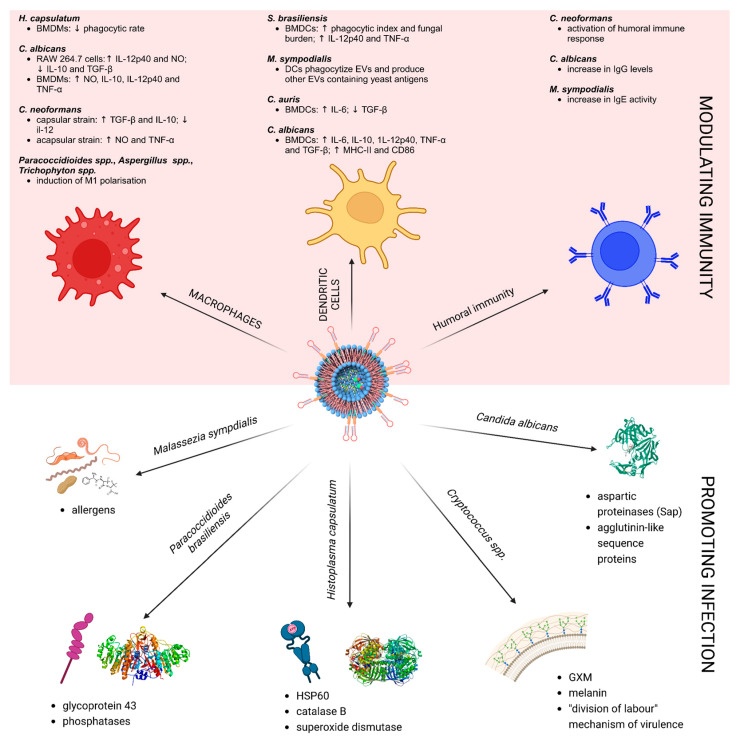
Immunomodulatory properties of fungal EVs (upper half) and their virulence factors (bottom half). Fungal EVs showed to exert immunomodulatory effects on mammalian macrophages [29,30,31,33,124,188,189], dendritic cells [32,67,124,190] and humoral immunity cells [65,190,191]. At the same time, they are known to promoting infections through the delivery of virulence factors, as seen in *M. sympodialis* [122], *P. brasiliensis* [119], *H. capsulatum* [74,127], *Cryptococcus* spp. [18,22,81,129,192,193] and *C. albicans* [194].

## Data Availability

Not applicable.

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
