# Peer review of "Immunomodulatory Potential of Fungal Extracellular Vesicles: Insights for Therapeutic Applications"

_biomolecules, 2023, doi:10.3390/biom13101487_

Round 1

Reviewer 1 Report

This was an extensive review on fungal EVs, and especially their composition, secretion, and ability to stimulate host immune responses.

In the final sentence of the abstract, the authors mention that engineered fungal cells could be used for the production of cargo- and antigen-specific EVs. But there is no clear indication in the review of what these engineered fungal cells would be, how they would be engineered, or for what purpose. What specific cargo? What specific antigens? The authors themselves state in their conclusion that much more research needs to be done in order to understand EV production, composition, and diversity. It seems far-fetched that engineered fungal EVs should be used in any situation when so much work remains to be done in understanding fungal EV production, composition, secretion, etc...

In the conclusion on line 640, the authors state that fungal EVs could be used to surmount biological barriers, including the blood-brain barrier. The paper they cite (ref. 254) to support this sentence discusses the use of self-derived mouse dendritic cells for exosome production. What would be the purpose of getting fungal EVs across the blood-brain barrier of a mammal? Would that not be dangerous?        

A few minor points:

1)  Maybe it was just because it was a proof, but none of the species names were italicized. That needs to be corrected.

2) The sentence starting on line 457 - "Taken together, these results suggested...) is awkward and difficult to understand. 

3) I found the top panel of Figure 2 difficult to understand. There are arrows pointing everywhere and there is no intuitive flow to it. The bottom panel is fine, but the top needs to be redesigned to better help the reader.  

There is no problem with the English. 

Reviewer 2 Report

The review is intriguing, convincing, and well-ritten, featuring an extensive array of citations. It is intresting and is the hot spot of the current research. However, the manuscript requires some issues to be discuss.

1. The abstrat need to be refined to stress the possible mechanism  of the immunomodulatory properties.

2.  Please rephrase lines 27-32.

3. Please provide any examples in lines 220-225 to illustrate the role of lipid components in EVs concerning membrane formation or drug resistance?
4. Kindly supplement Figure 1 with pertinent content for Section 2.4.
5. The content presented in lines 268-274 duplicates the material found in Section 2.1.
6. Please assess whether Section 3 adequately addresses both the characteristics and functions of EVs.

Minor editing of English language required
